# Effectiveness of Multimodal imaging for the Evaluation of Retinal oedema And new vesseLs in Diabetic retinopathy (EMERALD)

Noemi Lois,[1] Jonathan Cook,[2] Stephen Aldington,[3] Norman Waugh,[4] Hema Mistry,[4] William Sones,[5] Danny McAuley,[6] Tariq Aslam,[7] Claire Bailey,[8] Victor Chong,[9] Faruque Ghanchi,[10] Peter Scanlon,[11] Sobha Sivaprasad,[12] David Steel,[13,14] Caroline Styles,[15] Christine McNally,[16] Rachael Rice,[16] Lindsay Prior,[17] Augusto Azuara-Blanco,[18] On behalf of the EMERALD Study Group

For numbered affiliations see end of article.

**Correspondence to**
Prof. Noemi Lois;
n.lois@qub.ac.uk

## ABSTRACT

**Introduction** Diabetic macular oedema (DMO) and proliferative diabetic retinopathy (PDR) are the major causes of sight loss in people with diabetes. Due to the increased prevalence of diabetes, the workload related to these complications is increasing making it difficult for Hospital Eye Services (HSE) to meet demands.

**Methods and analysis** Effectiveness of Multimodal imaging for the Evaluation of Retinal oedema And new vesseLs in Diabetic retinopathy (EMERALD) is a prospective, case-referent, cross-sectional diagnostic study. It aims at determining the diagnostic performance, cost-effectiveness and acceptability of a new form of surveillance for people with stable DMO and/or PDR, which entails multimodal imaging and image review by an ophthalmic grader, using the current standard of care (evaluation of patients in clinic by an ophthalmologist) as the reference standard. If safe, cost-effective and acceptable, this pathway could help HES by freeing ophthalmologist time. The primary outcome of EMERALD is sensitivity of the new surveillance pathway in detecting active DMO/PDR. Secondary outcomes include specificity, agreement between new and the standard care pathway, positive and negative likelihood ratios, cost-effectiveness, acceptability, proportion of patients requiring subsequent full clinical assessment, unable to undergo imaging, with inadequate quality images or indeterminate findings.

**Ethics and dissemination** Ethical approval was obtained for this study from the Office for Research Ethics Committees Northern Ireland (reference 17/NI/0124). Study results will be published as a Health Technology Assessment monograph, in peer-reviewed national and international journals and presented at national/international conferences and to patient groups.

**Trial registration number** NCT03490318 and ISRCTN:10856638.

## INTRODUCTION

Diabetic retinopathy (DR) is the most common microvascular complication of diabetes and a leading cause of visual loss

### Strengths and limitations of this study

► Prospective, multicentre study.
► Masking of graders to results of the reference standard and masking of ophthalmologists to results of the grader's pathway.
► Evaluation of the acceptability of the new pathway to patients and health professionals.
► Early involvement in all aspects of the study of patients and public.
► Lack of fluorescein angiography to confirm active proliferative diabetic retinopathy.

among individuals of working age.[1] Patients with DR may lose sight as a result of the development of diabetic macular oedema (DMO) and/or proliferative diabetic retinopathy (PDR), the major complications of DR. In the former, fluid accumulates in the central part of the retina, the macula, which is responsible for detailed central vision. In the latter, abnormal new blood vessels ('new vessels') grow on the optic nerve head or on the surface of the retina and towards the inside of the eye (the vitreous cavity) leading to sight loss from haemorrhaging and/or traction on the retina with subsequent retinal detachment. Due to the increasing numbers of people with diabetes, it is expected that the burden of DR will continue to rise. Indeed, it has been estimated that the worldwide prevalence of DR will increase from 126.6 million in 2010 to 191 million by 2030.[2]

The prevalence of DMO in England was estimated to be 7% of the total diabetic population in 2010.[3] A very similar estimate of the prevalence was found in an individual participant data meta-analysis, which included 22 896 individuals from 35 studies, conducted in Asia,

BMJ

Australia, Europe and the USA, which provided an overall age-standardised prevalence of DMO of 6.8%.[4] Based on this published prevalence of DMO and considering the prevalence of diabetes in the UK,[5] it can be estimated that there are 280 000 people in the UK affected by DMO. In the UK, patients with DMO are treated with focal or grid macular laser (when the central retinal thickness (CRT), measured by means of spectral domain optical coherence tomography (SD-OCT), is <400 microns), which is delivered in a single session, or with injections into the eye of antivascular endothelial growth factor (anti-VEGF) therapies, currently in the National Health Service (NHS) ranibizumab (Lucentis) and aflibercept (Eylea) (when the CRT measured by SD-OCT is ≥400 microns).[6 7] Intraocular steroids are also available for patients who do not respond to the above therapies and are pseudophakic (ie, have had their cataracts removed).[8] Once treated, long-term follow-up is required to determine whether DMO recurs. Typically, patients are followed every 3–4 months following laser treatment for DMO or monthly initially and then every 1–3 months thereafter following treatment with anti-VEGFs.[9] Follow-up continues for the rest of the patient's life.

The estimated prevalence of PDR in the individual participant data meta-analysis referred to above[4] was 6.96%; based on the prevalence of diabetes in the UK,[5] there are ~278 400 people in the UK affected by PDR. At present in the UK, patients with PDR are treated with panretinal laser photocoagulation (PRP), which is delivered most often in two to three sessions. Once treatment is completed, patients are followed at 4–6 months intervals for their lifetime to determine whether reactivation occurs, as new vessels in PDR could come back and tractional retinal detachment can still ensue.[9] A high proportion of patients with DR followed in Hospital Eye Services (HES) have treated and inactive PDR.[10]

Currently in the NHS ophthalmologists with expertise in retinal diseases assess patients during follow-up visits. At each visit, patients with DMO are evaluated with a visual acuity test, most often undertaken by a nurse; SD-OCT, obtained by a photographer and interpreted by the ophthalmologist, and slit-lamp biomicroscopy, undertaken by an ophthalmologist, who then determines whether DMO is present or absent. SD-OCT is a non-invasive, user-friendly and safe imaging technique that obtains scans of the back of the eye. SD-OCT allows measurement of the CRT (which is often increased when DMO is present) and visualising fluid in the retina which is the hallmark of DMO. SD-OCT has been extensively used in clinical trials and clinical practice to determine the presence of DMO, select treatment and monitor the response to treatment.[11–17] In the follow-up of patients with PDR, ophthalmologists typically examine the patient by slit-lamp biomicroscopy. Photographs are not routinely obtained in clinic to determine whether or not active PDR is present. Standard cameras are not able to image the whole retina. It is possible, however, to obtain several images to capture the appearance of the centre, superior and inferior parts of the retina; the retinal periphery cannot be imaged with standard cameras. In recent years, new ultra-wide field imaging has become available, allowing imaging of the entire retina. This technology may be, thus, preferable, and it may be easier for patients and reduce the time required to obtain images.

Given the high number of people with DMO and PDR, the need for patients to be seen at short follow-up intervals, the need for frequent treatments and the requirement for long-term follow-up, there is a very large workload in HES related to DMO/PDR. This is making it difficult for the NHS to cope with the demand, in particular, due to shortage of ophthalmologists. This is only expected to get worse given the increasing prevalence of diabetes. Identifying new ways of increasing the NHS capacity and efficiency without compromising quality of care would greatly benefit the NHS.

The purpose of EMERALD is to determine whether successfully treated patients with DMO/PDR could be followed without a face-to-face examination by an ophthalmologist. EMERALD will evaluate a new care pathway, which will include multimodal retinal imaging and separate image assessment by trained ophthalmic graders. This new pathway will be compared with the current standard care pathway which is, for DMO, having an ophthalmologist evaluating patients in clinic by slit-lamp biomicroscopy and with access to SD-OCT images and for PDR evaluating patients in clinic by slit-lamp biomicroscopy. EMERALD will determine how accurate the new pathway is at determining which patients have active or inactive disease when compared with the standard of care and its costs and acceptability by patients and health professionals.

## METHODS AND ANALYSIS
### Study design and setting
Prospective, diagnostic study of patients with DMO, PDR or both, who had been previously successfully treated and who, at the time of enrolment in the study, may have active or inactive disease (both are required to evaluate the diagnostic performance of the new pathway).

EMERALD has a case-referent cross-sectional diagnostic study design with both sampling (selection) of patients and data collection carried out prospectively.[18] This approach provides both a cost-efficient study design and low risk of bias in terms of diagnostic accuracy.[19]

EMERALD is set within specialised HES in the UK. All participating centres have extensive experience with the management of DMO and PDR.

### Patient and public involvement
A patient and public involvement (PPI) group was established early on in the study, at the stage of trial conception. The PPI group confirmed the research question was important to patients, the tests to be undertaken for the purpose of the study were adequate and feasible to be performed in the clinical setting and the outcomes

measured relevant to them. The PPI group provided essential input to the patient-related materials produced for the study, including the patient information sheet and consent form. Furthermore, patients' views on the pathways of care investigated will be evaluated specifically in the study (see 'Focus groups discussions: assessment of acceptability of the new care pathway' section). The PPI group will be actively involved in the dissemination of the results. Patients were not involved in the recruitment of participants in the study.

## Participants: eligibility criteria
### Inclusion criteria
Adults (18 years of age or older) with type 1 or 2 diabetes with previously successfully treated DMO and/or PDR in one or both eyes and in whom, at the time of enrolment in the study, DMO and/or PDR may be active or inactive. Patients can be recruited only once.

Active DMO is defined as CRT of >300 microns and/or presence of intraretinal/subretinal fluid on SD-OCT due to DMO.

Inactive DMO is defined as no intraretinal/subretinal fluid.

Active PDR is defined by the presence of subhyaloid/vitreous haemorrhage and/or active new vessels (new vessels with lack of fibrosis on them).

Inactive PDR is defined by the lack of preretinal/vitreous haemorrhage and lack of active new vessels.

### Exclusion criteria
1. Unable to provide informed consent.
2. Patients who do not read, speak or understand English.

## Outcome measures
### Primary outcome
► Sensitivity of the new pathway (ophthalmic grader pathway) in detecting active DMO/PDR, using the standard care pathway as the reference standard.

### Secondary outcomes
► Specificity, concordance (agreement) between new pathway (ophthalmic grader pathway) and the standard care pathway, positive and negative likelihood ratios.
► Cost-effectiveness.
► Acceptability.
► Proportion of patients requiring subsequent full clinical assessment.
► Proportions of patients unable to undergo imaging, with inadequate quality images or indeterminate findings.

## Study procedures and schedule of assessments
### Ophthalmic grader pathway and standard care pathway
In EMERALD, a new clinical care pathway (figure 1), the ophthalmic grader pathway, will be evaluated and compared with the current standard of care.

If this new pathway were to be used in clinical practice, patients would undergo multimodal retinal imaging with

**Ophthalmic Grader Pathway**
- Images will be obtained from both eyes by an ophthalmic photographer/imaging technician, including:
  - Spectral Domain Optical Coherence Tomography (SD-OCT)
  - Early Treatment Diabetic Retinopathy Study (ETDRS) 7 field non-stereoscopic fundus photographs
  - Ultra-wide angle fundus images
- Images will be read by trained ophthalmic graders
  - Presence/absence of active DMO/PDR determined

**Figure 1** Summary of ophthalmic grader pathway. DMO, diabetic macular oedema; PDR, proliferative diabetic retinopathy; SD-OCT, spectral domain optical coherence tomography.

subsequent review by trained ophthalmic graders. If active disease were to be detected or if the ophthalmic grader were to be uncertain as to whether or not there was active DMO/PDR present, the patient would be referred to an ophthalmologist for a full assessment. If the patient were to be stable (ie, inactive DMO/PDR), the patient would remain under this model of surveillance with a predetermined interval.

For the purpose of EMERALD, all patients will go through the standard care pathway which is: 1) for DMO: ophthalmologist evaluating patients in clinic by slit-lamp biomicroscopy and with access to SD-OCT images, which are routinely obtained for the evaluation of DMO in clinic; 2) for PDR: ophthalmologists evaluating patients in clinic by slit-lamp biomicroscopy.

### Selection of ophthalmic graders and training
Currently, ophthalmic photographers/imaging technicians obtain images and interpret them routinely, but make no decisions on the care of patients. In ophthalmic services, there are also ophthalmic graders that have been trained to interpret findings on fundus images for the purpose of undertaking DR screening.

For EMERALD, the ophthalmic graders at each participating site will be selected as follows. First, local principal investigators (PIs) would provide names of individuals they believe have experience obtaining and/or grading images of patients with DMO and PDR; these individuals would be also confirming their interest and willingness to participate in EMERALD. It is possible that some ophthalmic graders selected for EMERALD will be already involved in the grading of images for DR screening.

Graders identified by the PIs will be asked to fill out questionnaires detailing their experience imaging/grading DMO/PDR, recognising features of DMO/PDR and whether they feel confident identifying DMO on SD-OCT and new vessels on fundus images. Graders stating they do not have experience imaging/grading DMO/PDR and/or those stating they could not recognise features of DMO/PDR will not be invited to take part in EMERALD as graders.

Formal training, during a training meeting(s) in which features of active/inactive DMO/PDR will be reviewed and discussed and where extensive clinical examples will be presented, will be provided to all EMERALD

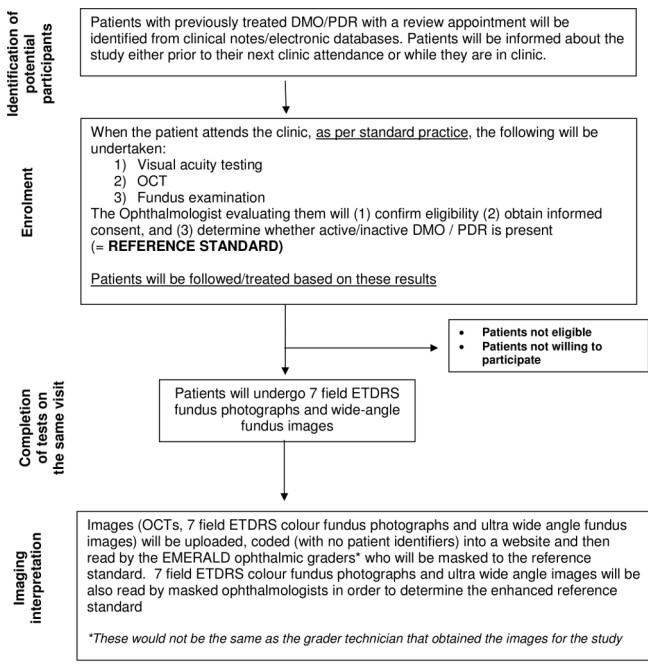

**Identification of potential participants**

Patients with previously treated DMO/PDR with a review appointment will be identified from clinical notes/electronic databases. Patients will be informed about the study either prior to their next clinic attendance or while they are in clinic.

**Enrolment**

When the patient attends the clinic, as per standard practice, the following will be undertaken:
1) Visual acuity testing
2) OCT
3) Fundus examination
The Ophthalmologist evaluating them will (1) confirm eligibility (2) obtain informed consent, and (3) determine whether active/inactive DMO / PDR is present
(= **REFERENCE STANDARD**)

Patients will be followed/treated based on these results

- Patients not eligible
- Patients not willing to participate

**Completion of tests on the same visit**

Patients will undergo 7 field ETDRS fundus photographs and wide-angle fundus images

**Imaging interpretation**

Images (OCTs, 7 field ETDRS colour fundus photographs and ultra wide angle fundus images) will be uploaded, coded (with no patient identifiers) into a website and then read by the EMERALD ophthalmic graders* who will be masked to the reference standard. 7 field ETDRS colour fundus photographs and ultra wide angle images will be also read by masked ophthalmologists in order to determine the enhanced reference standard

*These would not be the same as the grader technician that obtained the images for the study

DMO = diabetic macular oedema; PDR = proliferative diabetic retinopathy; OCT = optical coherence tomography; ETDRS = early treatment diabetic retinopathy study

**Figure 2** Study flow chart. DMO, diabetic macular oedema; ETDRS, early treatment diabetic retinopaty study; PDR, proliferative diabetic retinopathy; OCT, optical coherence tomography.

ophthalmic graders prior to the initiation of the study, as described in the EMERALD Study Manual.

To ensure graders selected will be in a position to undertake the task of grading images, all potential graders will be required to take a test involving the reading of OCTs and fundus images. Only those graders who reach a minimum of 80% of correct answers to detect presence of DMO or active PDR, when present, will be invited to act as graders. Graders will be allowed to undergo further training and take the test a second time but if the minimum of correct answers is not reached they will not be able to act as graders.

A web-based teaching module with examples of DMO/PDR will be prepared so that graders can access it to consolidate their knowledge. Clear guidelines on when patients would need to be referred for an assessment by an ophthalmologist will be also given.

### Schedule of assessments for EMERALD

Patient flow and procedures undertaken in EMERALD have been summarised in figure 2.

Written informed consent will be obtained by ophthalmologists/deputes from all participants prior to any study procedures being undertaken.

The following information obtained during the standard care pathway will be recorded in the case report forms (CRFs):
- Medical and ophthalmic history.
- Best-corrected visual acuity.

- Whether there was active or inactive DMO/PDR and whether or not there were features that could make imaging difficult (eg, small pupil).
- Details on presence/absence and location of active/inactive new vessels in the disc and elsewhere in the retina and/or preretinal haemorrhage/vitreous haemorrhage.
- Presence of any other co-existent eye disease noted.
- Information on the proposed plan for the patient.

Patients will undergo ETDRS 7-field and ultra-wide field imaging, obtained following the guidelines set in the EMERALD Study Manual and will complete EQ-5D 5L, NEI VFQ-25 and VisQoL questionnaires.

Patients willing to participate in the Focus Group Discussions will be consented; a group of them will be contacted at a later date for this purpose (see below).

Fundus photographs and SD-OCTs will be anonymised and transferred to Queen's University Belfast reading centre where they will be uploaded in an electronic website developed for the study; these anonymised images will be then made accessible to the ophthalmic graders (and ophthalmologists determining the 'enhanced reference standard' for PDR, see below) for the purpose of grading.

Ophthalmic graders will determine (and record in the appropriate CRF):
- Whether there is active/inactive DMO/PDR (or if unsure);
- Location of new vessels;
- Whether patient could continue review in the ophthalmic grader pathway or requires assessment by an ophthalmologist and the reasons why.

### Enhanced reference standard

The reference standard for PDR (ophthalmologist evaluating patients in clinic by slit-lamp biomicroscopy) will be used for the primary analysis. The reference standard for PDR, however, could potentially be improved. There is a possibility that new vessels (indicative of PDR) may not be seen by the ophthalmologist on slit-lamp biomicroscopy but could be detected in a fundus photograph. In order to determine the impact of this potential event, EMERALD will also evaluate an alternative 'enhanced' reference standard which will consist of the ophthalmologist assessment supplemented by the evaluation of the fundus images (7-field ETDRS and ultra-wide field fundus images) done by an ophthalmologist. This reading by the ophthalmologist of the fundus images will be done only after the reference standard has been set, to ensure it will not affect or influence the reference standard. If either, slit-lamp biomicroscopy, 7-field ETDRS or ultra-wide angle fundus images detect active PDR, the patient will be considered to have 'active' PDR under this 'enhanced' reference standard. This information will be recorded in the appropriate CRF. The PDR status based on the enhanced reference standard will be used in a sensitivity analysis of the new pathway's diagnostic accuracy.

## Masking

Ophthalmic graders will be masked to the reference standard; to ensure this, they will not interpret images from patients recruited at their own centre and will not have access to results of the reference standard. They will not read 7-field ETDRS and wide-angle fundus images or SD-OCTs of the same eye either, to ensure their reading of one imaging technology will not influence their reading of the other.

Ophthalmologists doing the standard of care evaluation will also be masked to the findings/decisions made by the ophthalmic graders (who will be reviewing the images at a later date).

Ophthalmologists determining the 'enhanced' reference standard will be masked in the same manner as the ophthalmic graders.

Patients will also be masked to findings/decisions made by the ophthalmic graders (these will not be available at the time of the study's clinical visit).

## Data collection and quality checks

CRFs will be used to collect data. Monitoring during the study will check the accuracy of entries on CRF's against source documents, adherence to protocol, procedures and to the International Conference of Harmonisation Good Clinical Practice guidelines and regulatory requirements. Monitoring visits will be undertaken by a monitor from the Northern Ireland Clinical Trials Unit (NICTU). To ensure accurate, complete and reliable data are collected, the Chief Investigator and the NICTU will provide training to site staff.

Study data will be entered onto a web-based Clinical Trial Database by NICTU personnel and processed electronically. Data quality control checks will be carried out by a data manager to ensure accuracy; data errors will be documented in Quality Control Reports with corrective actions implemented. Data validation will be implemented and discrepancy reports will be generated following data entry to identify discrepancies such as out of range, inconsistencies or protocol deviations based on data validation checks programmed in the clinical trial database.

## Sample size

The sample size was determined on the basis of setting a target of the number of reactivated (active) DMO and PDR patients, which would enable sensitivity to be tested against a prespecified target level of 80%.[20] This level was considered the minimum acceptable level for the new pathway (ophthalmic grader pathway) to be clinically viable. A lower specificity is considered acceptable and a target of 65% for specificity was used to confirm sufficiency of the sample size for assessing specificity. To detect the sensitivity of the new pathway (photographer/imaging technician pathway) with 80% and 90% power (10% and 12% higher than the 80% minimal target set) would require 89 participants with each DMO/PDR who have reactivated (active DMO/PDR), with two-sided 5%

significance level.[21] Ninety-three participants who have not reactivated would enable a specificity 15% (10%) higher than the 65% target to be detected with 90% power. A 95% CI for photographer sensitivity and specificity would have a CI (Wilson method) with a width of 10%–20% depending on the observed level.[22] Allowing for 10% missing/indeterminate results, 104 individuals who have re-activated and 104 who have not, are required (208 for each, DMO and PDR) which leads to a need for a maximum of 416 participants in the study overall; some participants may have both DMO and PDR contributing to both DMO and PDR targets.

## Data analysis plan

Outcomes for DMO and PDR will be assessed in two separate analyses. Participants will be categorised as having active, inactive or no DMO/PDR according to the diagnosis established at the standard care pathway at the person level (ie, using data from both eyes where appropriate). Active and inactive DMO/PDR will be subcategorised as previously successfully treated or not. Those which had previously successfully treated DMO/PDR constitute 'eligible' patients for the new pathway. This person-based assessment reflects the consequences of the clinical decision. The diagnostic performance of the new pathway will be quantified by assessing against the standard care pathway. Reflecting how the new pathway would function in practice, an 'unsure' classification or an 'active' classification will both require an examination by an ophthalmologist under the main analyses.

A number of sensitivity analyses will be carried out. These will include 1) an assessment of the impact of the 'unsure' ophthalmic classification on the diagnostic performance; 2) using the ophthalmologist's decision to do further treatment rather than presence of active disease, given that in some patients with active disease the ophthalmologist may still consider observation if only very mild disease is present, 3) detection of disease in more severe cases (eg, high-risk PDR) and 4) the diagnostic performance within routine NHS clinics versus those set-up specifically for this research study. In addition, for PDR, a sensitivity analysis will assess the diagnostic performance of the ophthalmic grader against the 'enhanced' reference standard.

The impact of using ultra-wide field or 7-field ETDRS images on the diagnostic performance of the new pathway will be assessed also under the principal analyses for PDR using both the standard and the enhanced reference standard.

Furthermore, secondary analyses will include 1) an evaluation based on eye level data, 2) an analysis that will include all patients (ie, including those with no disease), 3) an assessment of using tests of both DMO and PDR with regard to an overall referral status and 4) the additional use of visual acuity as a proxy to detect active disease. We will quantify the proportion of patients identified by an ophthalmologist as having other eye comorbidities (eg,

epiretinal membrane, glaucoma, others) and will explore in relevant analyses their influence on the findings.

Sensitivity, specificity, positive and negative likelihood ratios will be calculated (with appropriate 95% CIs) for the alternative strategy using the current standard of care pathway findings as the reference standard. Agreement (concordance) between the new and current standard of care pathways will also be calculated (with 95% Wilson CI).[22] The difference in sensitivity and specificity between ultra-wide field and 7-field ETDRS images assessed by the ophthalmic graders will be compared with corresponding 95% CIs produced using Newcombe's method for paired data.[23]

The proportion of patients requiring subsequent full clinical assessment or unable to undergo assessments, with inadequate quality images or indeterminate findings will be calculated for the alternative pathway with corresponding CIs. All analyses will be carried out using STATA V.15.[24]

A detailed statistical analysis plan will be written by the trial statistician prior to the final analysis.

## Health economic analysis

This analysis will take into account: 1) sensitivity and specificity of the ophthalmic grader pathway for both DMO and PDR; 2) whether this new pathway detects any PDR missed by the current standard of care and 3) relative costs.

In the ophthalmic grader pathway, patients will be in one of four groups depending on the decisions made by these staff after reading the images: 1) true negative—no treatment required and patient will return for follow-up at the usual interval; 2) true positive—referred for ophthalmic assessment and treatment as required; 3) false negative—patient who may come to harm by visual loss; 4) false positive—patient will be referred to the ophthalmologist but will not require treatment; these patients will not come to harm apart from possible anxiety and inconvenience, but will consume ophthalmologist time.

Scores obtained in the health-related quality of life questionnaire (EQ-5D-5L) and visual function questionnaires (NEI VFQ-25 and Vis-QoL) will provide utility data for different health states. Resource use data will be collected to explore the costs of delivering the standard care pathway and the ophthalmic grader pathway and to find the key cost drivers. This will mainly consist of staff costs. Costs of 7-field ETDRS images will be compared with those of ultra-wide angle imaging.

If the sensitivity and specificity of the new and standard pathways were exactly the same there would be no quality-adjusted life year (QALY) differences, although there might be some disutility from process changes (ie, if one pathway caused more anxiety than the other). The key gain would be ophthalmologist time freed for other activities. The real benefits might be reduction in waiting times and earlier treatment of other patients leading to QALY gains for them. Such benefits would be difficult to estimate and the simplest measure would be ophthalmologist

sessions/days released for other activities. However, we will identify ways in which time released would be used. An underlying assumption to be checked is that the cost of assessment by the ophthalmic graders is less than that of the ophthalmologists. Both pathways would require nurse or optometrist time for checking visual acuity, as done in routine clinical practice. The costs of the image-based pathways will include both time for taking and reading images, using both conventional and ultra-wide field cameras. If there was marginal loss in sensitivity from the new pathway the consequences could be visual loss before next visit was due, or detection at next visit (with or without visual loss occurring) followed by possibly later than optimal treatment. Both could have disutilities.

It should be noted that assessment in clinical practice is repeated over time so active disease missed at one assessment might be picked up at the next. Given the cross-sectional design of EMERALD and the fact that all patients will undergo the standard care pathway, we will not be able to assess the disutility of any visual changes in patients recruited.

We will develop a Markov model using data from both this study (the EQ-5D 5L data for different states) and published studies on progression, so analysis of the effect of a reduced sensitivity will include, for DMO and PDR separately: 1) probability of progression before next visit, 2) probability that this would lead to irreversible visual loss and if so, how much; 3) if there was irreversible visual loss, the resulting disutility and QALY loss.

Specificity would be the determining factor in savings in ophthalmologist time: the poorer the specificity the lower the savings. In the PDR group, ultra-wide field imaging will be compared with both, standard care (ophthalmologist evaluating patients by slit-lamp biomicroscopy) and 7-field ETDRS to assess the cost-effectiveness of this more recently introduced imaging modality.

Any future costs and benefits will be discounted at 3.5%. NHS and personal social services perspective will be adopted. The model will be populated by cost, sensitivity and specificity data from the study and by estimates of progression, effectiveness of treatment (prompt and delayed), quality of life and future costs from published literature and expert opinion. Results will be expressed as cost per QALY gained. Appropriate sensitivity analyses will be conducted to assess robustness of results. Probabilistic sensitivity analyses will be undertaken to explore uncertainty in model parameters and to allow presentation of cost-effectiveness acceptability curves.

## Focus groups discussions: assessment of acceptability of the new care pathway

The acceptability of the ophthalmic graders pathway will be evaluated through focus group discussions. Patients' views on the acceptability of the proposed new pathway are essential if this were to be incorporated into clinical practice. Focus group discussions are particularly useful to help identify issues that resonate with lay people in matters of healthcare and have been widely used in

health services research; focus group discussions reach the parts that other methods cannot.[25] Their use in this study will enable us to acquire data on a full range of issues—some of which are unlikely to have been anticipated by professionals. The sample frame is designed to include consenting participants drawn from different UK areas and different age groups. Group discussions will be facilitated by a trained researcher. Discussions will be audio-recorded and later transcribed for analysis.

Informed consent will be obtained from participants prior to the focus group discussions. Participants consenting to take part will be approached at a later date via letter/phone call to inform them about the date/location/time of the meeting.

EMERALD will also examine the acceptability of the new pathway to health professionals. For this purpose, a small number of focus groups will be conducted involving photographer/imaging technicians/graders and ophthalmologists, all recruited from staff at participating sites.

The data from the focus groups will be analysed by the use of simple content analysis strategies. The focus of the analysis will be on 'acceptability' of the new alternative pathway and factors that might facilitate or impede such acceptability.[26]

## ETHICS AND DISSEMINATION

The study is conducted in compliance with the protocol approved by the Research Ethics Committee (REC). Changes to the protocol may require REC approval prior to implementation. The NICTU in collaboration with the sponsor will submit all protocol modifications to the REC for review in accordance with the governing regulations.

Serious adverse events will be recorded and reported to the Office for Research Ethics Committees Northern Ireland.

In order to maintain participant's confidentiality, a unique study identification number will be used for each participant. Study reports and communication regarding the study will identify the patients by their assigned unique trial identifier. Computers where information will be stored will be password protected.

A report containing the methodology and results of EMERALD will be published as a Health Technology Assessment monograph, freely accessible via the National Institute for Health Research (NIHR) Health Technology Assessment (HTA) website. The Royal College of Ophthalmologist will be contacted when the study is completed to allow the study findings to be incorporated in future diabetic retinopathy guidelines. Findings will be published in national/international peer-reviewed journals and presented at national/international meetings and patient groups.

## Author affiliations
[1]Ophthalmology, Wellcome-Wolfson Institute for Experimental Medicine, Queen's University, Belfast, UK
[2]NDORMS, University of Oxford, Oxford, UK
[3]Gloucester Hospitals NHS Foundation Trust, Gloucester, UK
[4]The Division of Health Sciences, University of Warwick, Warwick, UK
[5]NDORMS, University of Oxford, Oxford, UK
[6]Wellcome-Wolfson Institute for Experimental Medicine, Queens University, Belfast, UK
[7]The Manchester Academic Health Science Centre, University of Manchester, Manchester, UK
[8]Ophthalmology Department, Bristol Eye Hospital, Bristol, UK
[9]Ophthalmology, Royal Free Hospital NHS Foundation Trust, London, UK
[10]Ophthalmology, Bradford Royal Infirmary, Bradford, West Yorkshire, UK
[11]Gloucestershire Hospitals NHS Foundation Trust, Cheltenham, Gloucestershire, UK
[12]NIHR Moorfields Biomedical Research Centre, Moorfields Eye Hospital, London, London
[13]Sunderland Eye Infirmary, Sunderland, UK
[14]Institute of Genetic Medicine, Newcastle University, Newcastle upon Tyne, UK
[15]Ophthalmology, Queen Margaret Hospital, Fife, UK
[16]Northern Ireland Clinical Trials Unit, Belfast, UK
[17]Queen's University Belfast, Belfast, UK
[18]Centre for Public Health, Queen's University, Belfast, UK

**Acknowledgements** The authors would like to thank the optometrists, research coordinators, research nurses, ophthalmic technicians and ophthalmic photographers for contributing to EMERALD at each of the participating sites. The authors would like to thank all the patients participating in EMERALD, those who were part of the PPI group and Diabetes UK, Northern Ireland. The authors would like to thank Janice Bailie, Alison Murphy and Lynn Murphy for their support, help and assistance with this diagnostic study and the UK Ophthalmology Clinical Research Network, Maurice O'Kane, Julie Silvestri, Jonathan Jackson, Paul Biagioni and the Ophthalmology Clinical Research Network for their support to EMERALD.

**Collaborators** EMERALD Study Group: Ahmed Saad, James Cook University Hospital, South Tees Hospitals NHS Foundation Trust; Augusto Azuara-Blanco, Queen's University and Royal Victoria Hospital, Belfast H&SC Trust; Caroline Styles, Queen's Margaret Hospital, Fife; Christine McNally, Andrew Jackson and Rachael Rice, Northern Ireland Clinical Trials Unit; Clare Bailey, Bristol Eye Hospital, University Hospitals Bristol NHS Foundation Trust; Danny McAuley, Queen's University and Royal Victoria Hospital, Belfast H&SC Trust; David H Steel, Sunderland Eye Infirmary, City Hospitals Sunderland NHS Foundation Trust; Faruque D Ghanchi, Bradford Teaching Hospitals NHS Trust; Geeta Menon, Frimley Park Hospital NHS Foundation Trust; Haralabos Eleftheriadis, Kings College Hospital NHS Foundation Trust; Hema Mistry, Warwick University; Jonathan Cook and William Sones, Centre for Statistics in Medicine, University of Oxford; Lindsay Prior, Centre for Public Health, Queens University, Belfast; Nachiketa Acharya, Sheffield Teaching Hospitals NHS Foundation Trust; Noemi Lois, Queen's University and Royal Victoria Hospital Belfast H&SC Trust; Norman Waugh, Warwick University; Rachael Rice, Northern Ireland Clinical Trials Unit; Samia Fatum, John Radcliffe Hospital, Oxford University Hospitals NHS Foundation Trust; Sobha Sivaprasad, Moorfields Eye Hospital NHS Foundation Trust; Stephen Aldington and Peter H Scanlon, Gloucestershire Hospitals NHS Foundation Trust; Tariq M Aslam, Manchester Royal Eye Hospital, Central Manchester University Hospitals NHS Foundation Trust; Victor Chong, Royal Free Hospital NHS Foundation Trust, London. Clinical sites participating in recruitment: The Belfast Health and Social Care Trust, Belfast, Northern Ireland; Bradford Royal Infirmary, Bradford Teaching Hospitals NHS Foundation Trust; Bristol Eye Hospital, University Hospitals Bristol NHS Foundation Trust; Frimley Park Hospital NHS Foundation Trust; Gloucestershire Royal Hospital, Gloucestershire Hospitals NHSF Trust; James Cook University Hospital, South Tees Hospitals NHS Foundation Trust;Kings College Hospital NHS Foundation Trust; Manchester Royal Eye Hospital, Central Manchester University Hospitals NHS Foundation Trust; Moorfields Eye Hospital NHS Foundation Trust; John Radcliffe Hospital, Oxford University Hospitals NHS Foundation Trust; Queen Margaret Hospital, Fife; Sheffield Teaching Hospitals NHS Foundation Trust; Sunderland Eye Infirmary, City Hospitals Sunderland NHS Foundation Trust. EMERALD is currently recruiting. The first trial participant was recruited in October 2017. Trial Management Group: Professor Noemi Lois (Chief Investigator), Professor Augusto Azuara-Blanco, Mr Steve Aldington, Dr Danny McAuley, Mr Peter Scanlon, Mr Lindsay Prior, Mrs Clare Newall, Mrs Michelle McGaughey, Mrs Christine McNally, Miss Rachael Rice, Mr Andrew Jackson, Mr Jonathan Cook, Mr William Sones, Professor Norman Waugh, Dr Hema Mistry, Mr Mark Wilson, Mrs Nuala Hannaway, Mrs Catherine Campbell. Trial Steering Committee: Mr John Norrie (Chair); Mr David Owens; Mrs Florence Findlay-White; Mr Winfried Amoaku; Miss Yemisi Takwoingi. The roles and responsibilities of the TSC can be found in https://www.nihr.ac.uk/funding-and-support/documents/funding-for-research-studies/how-to-apply/NETSCC_Project_Oversight_Groups_Guidance.pdf. A study Data and Ethics Monitoring Committee was not deemed

necessary given the extremely low risk of this diagnostic study. Trial Sponsor: The Belfast Health and Social Care Trust, Belfast, Northern Ireland. Contact person: Ms Alison Murphy; alison.murphy@belfasttrust.hscni.net.

**Contributors** NL conceived the study and drafted the protocol which was refined with input from JC, SA, NW, HM, WS, DMcA, TA, CB, VC, FG, PS, SS, DS, CS, LP and AA-B; JC and NL determined the sample size for the study and JC, WS and NL determined the statistical analysis plan. NW and HM planned the cost-effectiveness evaluation. LP designed and planned the focus group discussions and will carry them out. CM, RR and AJ provided management coordination and data management to the study. NL, AS, CB, DS, FG, GM, HE, NA, SF, SS, SA, PHS, TA will recruit and evaluate patients for the study. All authors reviewed the manuscript, provided input to it and approved the final submitted version.

**Funding** This project was funded by the NIHR (Health Technology Assessment programme) (project number 13/142/04).

**Disclaimer** The views expressed are those of the authors and not necessarily those of the NHS, the NIHR or the Department of Health and Social Care. Neither the Sponsor nor the Funder had any role on the study design; collection, management, analysis and interpretation of data; writing of this manuscript or in the decision to submit this manuscript for publication.

**Competing interests** NL, NW, AA-B, HM, DMcA, TA: none; CB: has been ad hoc advisor for Alcon, Bayer, Novartis, Alimera Sciences and Allergan; VC is a part-time employee of Boehringer Ingelheim International GmBH (BII), Germany; this study, however, is not being undertaken as part of the employment with BII and, thus, the content of this manuscript is not endorsed by BII. VC has also received speaker fees from Quantel Medical, France; SS has received research grants, travel feeds and attended advisory board meetings of Novartis, Bayer, Roche, Allergan, Heidelberg Engineering, Optos and Boehringer Ingelheim; DS acted as consultant to Alcon, attended advisory boards for Novartis and Bayer and received research funding from Bayer and Alcon. PS has attended Advisory Boards for Allergan, Roche, Boehringer and Bayer; his department has received Educational, Research or Audit Grants from Allergan, Novartis and Bayer.

**Patient consent for publication** Not required.

**Ethics approval** Ethics approval was obtained from the Office for Research Ethics Committees Northern Ireland (ORECNI-17/NI/0124).

**Provenance and peer review** Not commissioned; externally peer reviewed.

**Author note** Authorship policy: An author is considered someone who has made a substantive intellectual contribution to the study and the relevant report.

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
