## [Reviewer comments · BMJ Open]

ARTICLE DETAILS

TITLE (PROVISIONAL)	Effectiveness of Multimodal imaging for the Evaluation of Retinal oedema And new vessels in Diabetic retinopathy (EMERALD): Protocol for a Diagnostic Study
AUTHORS	Lois, Noemi; Cook, Jonathan; Aldington, Stephen; Waugh, Norman; Mistry, Hema; Sones, William; McAuley, Danny; Aslam, Tariq; Bailey, Claire; Chong, Victor; Ghanchi, Faruque; Scanlon, Peter; Sivaprasad, Sobha; Steel, David; Styles, Caroline; McNally, Christine; Rice, Rachael; Prior, Lindsay; Azuara-Blanco, Augusto

VERSION 1 - REVIEW

REVIEWER	Paul McKeigue UNIVERSITY OF EDINBURGH Professor of Genetic Epidemiology & Statistical Genetics
REVIEW RETURNED	05-Dec-2018

GENERAL COMMENTS	1 . Sensitivity as primary outcome does not make sense. It is always possible to achieve 100% sensitivity by setting a low enough threshold for classifying as test positive. To take both sensitivity and specificity into account, the C-statistic can be used. To evaluate the incremental contribution of an extra measurement, the expected information for discrimination (McKeigue 2018) can be used.2. The reliability of the standard grading by ophthalmologist using slit-lamp and OCT should be assessed, at least in a subset.3. The objective of reducing costs and workload of grading by ophthalmologists is clear enough, but it's surprising that the applicants are not evaluating automated grading of fundus photographs and OCT images at the same time as or even instead of manual grading. Deep learning algorithms have in the last ten years transformed the ability of machines to solve this sort of classification problem. Grading of fundus photographs for diabetic retinopathy is practically a solved problem, with C-statistics of 0.99. Development of deep learning for scoring of macular oedema in OCT images is still at an early stage, but results are promising. Deep learning requires very large training datasets, but these are available for retinal images in the public domain, and for OCT through collaboration.4. Use of automated grading would completely change the economics, as the marginal cost of grading is practically nil once
--

	the algorithm has been trained and implemented in a robust software package.
--	--

REVIEWER	maurizio battaglia parodi Ophthalmology Department, Vita-Salute San Raffaele University, Milan, Italy
REVIEW RETURNED	21-Feb-2019

GENERAL COMMENTS	I believe that this project is really useful to both patients and scientific community
--

REVIEWER	Wassef Chanbour Beirut eye and ENT specialist hospital-LEBANON
REVIEW RETURNED	02-Mar-2019

GENERAL COMMENTS	I would like to congratulate the authors for this protocol and I hope that the results could influence the screening of macular edema and proliferative retinopathy allowing more patients to be examined in the cities where there is a shortage in the number of ophthalmologists. Few changes needs to be performed before publication:  -In the introduction: each of the websites between brackets (line 36-39-44-49-56) should be given a number and cited as a reference in the references section instead of the introduction. - The objectives are well defined -the methods are clear. However, one topic needs to be more clarified by the authors. in order to generalize the use of the new screening method using ophthalmic graders, other confounding factors should be accounted for. (Page 7 of 30 Ophthalmic Grader Pathway and Standard Care Pathway) For example, it is not clear whether ophthalmic graders may have access to the patient's visual acuities. Other factors may cause a drop of vision in this population like the development of cataract that may not blur the retinal images and OCT. Many of these patients will have a delayed checkup by their ophthalmologist if the grader decides that there is no macular edema or PDR missing other diagnosis. In the study all the patients will be examined by an ophthalmologist using a slit lamp. This is why the abnormal slit lamp results (other than macular edema and PDR) should identified, reported and compared between both groups. Finally, the SPIRIT checklist needs to be adjusted: (the number on the lower right side of each page of the manuscript was used in the checklist)  14-sample size : page 9 17 a-b Blinding page 8 20-a-b-c Statistics: 10/11/12 24-Research ethics approval: 12 25-Protocol amendments: 12 26-a Consent or assent: 7 27- Confidentiality: 9-12 28- Dissemination policy: trial results: 12
--

VERSION 1 – AUTHOR RESPONSE

Reviewer(s)' Comments to Author:

Reviewer: 1

Reviewer Name: Paul McKeigue

Institution and Country: UNIVERSITY OF EDINBURGH, Professor of Genetic Epidemiology & Statistical Genetics

Please state any competing interests or state 'None declared': None declared

Please leave your comments for the authors below

1 . Sensitivity as primary outcome does not make sense. It is always possible to achieve 100% sensitivity by setting a low enough threshold for classifying as test positive. To take both sensitivity and specificity into account, the C-statistic can be used. To evaluate the incremental contribution of an extra measurement, the expected information for discrimination (McKeigue 2018) can be used.

We agree with Reviewer 1 in that the diagnostic performance of the new pathway cannot be judged on the basis of sensitivity alone. On this regard, it is worth mentioning that this project was reviewed by the National Institutes of Health Research (NIHR). Initially, we had a dual primary outcome, which included, as the Reviewer suggests, sensitivity and specificity. Following peer review of our grant, the NIHR Board requested this was changed and asked us to have only sensitivity as our primary outcome. They noted that sensitivity was more important than specificity in this setting; the new pathway may be still cost-effective and resource efficient even if the specificity were to be relatively low, but would be only acceptable if the sensitivity were to be high.

We will however, interpret, EMERALD findings based on both, sensitivity and specificity. In that regard it is not that different from any study where there is a stated primary outcome, but the secondary outcomes inform the overall interpretation of the results.

We have not added further information to the manuscript on this regard.

2. The reliability of the standard grading by ophthalmologist using slit-lamp and OCT should be assessed, at least in a subset.

The standard care pathway used in EMERALD is the current standard of care in the NHS. Thus, independently of what its reliability is, this had to be used as comparator in this study.

As mentioned, EMERALD is an NIHR funded study, which is already ongoing. Adding new components to the investigation is not possible at this stage. It would require ethical and local approval at all sites and NIHR review and approval too of any changes made. This addition would require also funding.

However, on the point raised by the Reviewer, we would like to mention that we will be determining an enhanced reference standard for PDR, as stated in the protocol. Patients will have an assessment of PDR based on the slit-lamp examination done by the ophthalmologist and also based on the assessment of images by the graders and by the ophthalmologists.

We have not added any further information to the manuscript on this matter.

3. The objective of reducing costs and workload of grading by ophthalmologists is clear enough, but it's surprising that the applicants are not evaluating automated grading of fundus photographs and OCT images at the same time as or even instead of manual grading. Deep learning algorithms have in the last ten years transformed the ability of machines to solve this sort of classification problem. Grading of fundus photographs for diabetic retinopathy is practically a solved problem, with C-statistics of 0.99. Development of deep learning for scoring of macular oedema in OCT images is still at an early stage, but results are promising. Deep learning requires very large training datasets, but these are available for retinal images in the public domain, and for OCT through collaboration.

We agree with the Reviewer that deep learning will likely substitute in many instances human grading in the future. However, at present, this is not the case. The type of eyes of patients included in EMERALD (i.e. those with proliferative diabetic retinopathy (PDR) and diabetic macular oedema (DMO) previously treated with anti-vascular endothelial growth factor (anti-VEGFs) therapies and laser) are more complex than naïve, untreated eyes. At present, there is no data suggesting that deep learning is able to discern accurately active new vessels on fundus images, even in naïve eyes. This is also the case, as the Reviewer already suggests, for DMO.

At this year's annual USA Macula Society Meeting, experts in deep learning technologies and imaging, including Prof Ursula Schmidt-Erfurth, Prof Michael Abramoff and Dr Daniel Ting, acknowledged that further work needs to be done before these technologies are ready to be used.

However, it is in our plan, once EMERALD is completed, to use the images produced in EMERALD, which would have been graded by graders and ophthalmologists, to undertake a study evaluating the diagnostic performance of using deep learning for the evaluation of eyes of patients with previously treated PDR and DMO, and we plan to seek further funding for this project from the NIHR.

We have not added further information on this matter to the manuscript.

4. Use of automated grading would completely change the economics, as the marginal cost of grading is practically nil once the algorithm has been trained and implemented in a robust software package.

As mentioned above, at present time automated grading for the group of patients enrolled in EMERALD is not possible. But we will certainly pursue this strategy in future studies and we thank the Reviewer for the suggestion.

We have not added further information to the manuscript on this regard.

Reviewer: 2

Reviewer Name: Maurizio Battaglia Parodi

Institution and Country: Ophthalmology Department, Vita-Salute San Raffaele University, Milan, Italy

Please state any competing interests or state 'None declared': None

Please leave your comments for the authors below

I believe that this project is really useful to both patients and scientific community.

We thank Reviewer 2 for his comments.

Reviewer: 3

Reviewer Name: Wassef Chanbour

Institution and Country: Beirut eye and ENT specialist hospital-LEBANON

Please state any competing interests or state 'None declared': none declared

Please leave your comments for the authors below

I would like to congratulate the authors for this protocol and I hope that the results could influence the screening of macular edema and proliferative retinopathy allowing more patients to be examined in the cities where there is a shortage in the number of ophthalmologists.

We thank Reviewer 3 for this comment.

Few changes needs to be performed before publication:

-In the introduction: each of the websites between brackets (line 36-39-44-49-56) should be given a number and cited as a reference in the references section instead of the introduction.

In order to address this issue (and that raised by the Editor) we have now given reference numbers to the website links and added these to the list of references.

- The objectives are well defined

Thank you very much.

-the methods are clear. However, one topic needs to be more clarified by the authors.

in order to generalize the use of the new screening method using ophthalmic graders, other confounding factors should be accounted for. (Page 7 of 30 Ophthalmic Grader Pathway and Standard Care Pathway). For example, it is not clear whether ophthalmic graders may have access to the patient's visual acuities. Other factors may cause a drop of vision in this population like the development of cataract that may blur the retinal images and OCT. Many of these patients will have a delayed check-up by their ophthalmologist if the grader decides that there is no macular edema or PDR missing other diagnosis.

We thank Reviewer 3 for this comment. Indeed, it would be appropriate for us to clarify this in the protocol.

In EMERALD we are recording visual acuity for all patients as well as presence of other eye co-morbidities, including cataracts, glaucoma or any others present. Graders do not have access though to visual acuity results when grading the images. Ophthalmologists evaluating patients and graders grading the images will be recording the presence of other eye co-morbidities noted.

In order to address these issues raised by the Reviewer and to further clarify the analysis we plan to undertake, we are providing now further details in the Data Analysis Plan, as follows:

"A number of sensitivity analyses will be carried out. These will include 1) an assessment of the impact of the "unsure" ophthalmic classification upon the diagnostic performance; 2) of using the ophthalmologist's decision to do further treatment rather than presence of active disease, given that in some patients with active disease the ophthalmologist may still consider observation if only very mild disease is present, 3) detection of disease in more severe cases (e.g. high risk PDR), and 4) of the diagnostic performance within routine NHS clinics versus those set-up specifically for this research study. In addition, for PDR, a sensitivity analysis will assess the diagnostic performance of the ophthalmic grader against the "enhanced" reference standard

The impact of using ultra-wide field or 7 field ETDRS images on the diagnostic performance of the new pathway will be assessed also under the principal analyses for PDR using both the standard and the enhanced reference standard.

Furthermore, secondary analyses will include 1) an evaluation based on eye level data, 2) an analysis that will include all patients (i.e. including those with no disease), 3) an assessment of using tests of both DMO and PDR with regards to an overall referral status, and 4) the additional use of visual acuity as a proxy to detect active disease. We will quantify the proportion of patients identified

by an ophthalmologist as having other eye comorbidities (e.g. epiretinal membrane, glaucoma, others) and will explore in relevant analyses their influence on the findings”.

In the study all the patients will be examined by an ophthalmologist using a slit lamp. This is why the abnormal slit lamp results (other than macular edema and PDR) should identified, reported and compared between both groups.

Yes, this is exactly what it being done in EMERALD. We thank the Reviewer, as mentioned above, for pointing this omission in the protocol. We have added, as stated above, information to the protocol on this regard.

Finally, the SPIRIT checklist needs to be adjusted: (the number on the lower right side of each page of the manuscript was used in the checklist)

We thank Reviewer 3 for noticing the need for us to adjust this.

We have now incorporated these changes to the SPIRIT checklist, as suggested below, by the Reviewer.

14-sample size : page 9

17 a-b Blinding page 8

20-a-b-c Statistics: 10/11/12

Statistics starts in page 9, so we have corrected it to 9-12

24-Research ethics approval: 12

25-Protocol amendments: 12

26-a Consent or assent: 7

27- Confidentiality: 9-12

Confidentiality is mentioned in page 12 so we have changed it to 12 (rather than 9-12)

28- Dissemination policy: trial results: 12